# “The Best Laid Plans”: Do Individual Differences in Planfulness Moderate Effects of Implementation Intention Interventions?

**DOI:** 10.3390/bs12020047

**Published:** 2022-02-14

**Authors:** Sabryna V. Sas, Kyra Hamilton, Martin S. Hagger

**Affiliations:** 1School of Applied Psychology, Mt Gravatt Campus, Griffith University, 176 Messines Ridge Road, Mt Gravatt, QLD 4122, Australia; sabryna.sas@griffith.edu.au (S.V.S.); kyra.hamilton@griffith.edu.au (K.H.); 2Menzies Health Institute Queensland, Gold Coast Campus, Griffith University, Southport, QLD 4215, Australia; 3Health Sciences Research Institute, University of California, Merced, 5200 N. Lake Road, Merced, CA 95343, USA; 4Department of Psychological Sciences, University of California, Merced, 5200 N. Lake Road, Merced, CA 95343, USA; 5Faculty of Sport and Health Sciences, University of Jyväskylä, FI-40014 Jyväskylä, Finland

**Keywords:** planfulness, planning, implementation intentions, intentions, personality, go no-go, reaction times

## Abstract

While there is good evidence supporting the positive effect of planning strategies like implementation intentions on the relationship between intention and behavior, there is less evidence on the moderating role of individual differences in planning capacity on this effect. This study aimed to examine the role of individual differences in planfulness on the effect of planning strategies on the intention–behavior gap. Specifically, this study investigated the influence of planfulness on the effectiveness of implementation intentions on goal-directed behavior using an experimental design. Undergraduate university students (*N* = 142) with high and low levels of planfulness based on a priori scores on a planfulness measure were randomized to either a planning (implementation intention) or familiarization (control) condition prior to completing a computerized go no-go task. We predicted that individuals reporting low levels of planfulness would be more effective in executing goal-directed behavior when prompted to form an implementation intention compared to individuals who do not receive a prompt. Additionally, we predicted that individuals reporting high planfulness would be equally effective in enacting goal-directed behaviour regardless of whether they formed an implementation intention. The results revealed no main or interaction effects of implementation intention manipulation and planfulness on task reaction times. The current results do not provide support for the moderating effect of planfulness on the implementation effect. The findings of this study were inconsistent with previous literature. This research has implications for the effectiveness of implementation intentions, as well as opportunities for further replication in a novel research area.

## 1. Introduction

Identifying the factors that influence individuals’ formation of goal intentions and engaging in action to attain the goal is important for achieving desirable outcomes such as reducing smoking, and increasing healthy eating and physical-activity participation [1,2,3,4]. However, intentions do not always lead to action and goal attainment; this finding has been labeled the intention–behavior “gap.” Based on dual-phase models of action, research has indicated that planning strategies such as implementation intentions may assist in strengthening the intention–behavior relationship and, ultimately, promote better goal attainment [5,6,7,8,9,10]. However, to date, there remains relatively little research exploring the effects of conditions within the individual that may facilitate or undermine the effectiveness of planning strategies on intention enactment and goal attainment.

A candidate individual difference construct that may moderate the intention–behavior relationship is planfulness [11]. Planfulness is an individual difference construct that captures an individual’s generalized tendency to adopt efficient goal-related cognitions in pursuit of their goals [11]. Research has suggested that individuals that exhibit good planfulness are more effective in achieving long-term goal progress [11], and a recent study in the physical-activity domain demonstrated a positive association between planfulness and gym attendance [12]. The present study aimed to investigate the extent to which planfulness moderates the effects of implementation intention interventions on goal-directed behavior. Based on previous theory and research, we supposed that individuals who lack a planning capacity would be the key beneficiaries of interventions that promote planning strategies like implementation intentions. Specifically, we predicted that individuals reporting low levels of planfulness would be more effective in executing goal-directed behavior when prompted to form an implementation intention relative to individuals who do not receive a prompt. Analogously, we predicted that individuals reporting high planfulness would be equally effective in enacting goal-directed behavior regardless of whether or not they formed an implementation intention.

### 1.1. The Intention–Behavior Relationship

Social cognition theories such as the theories of reasoned action and planned behavior (TPB) identify intention as the immediate antecedent of behavior [13,14,15,16]. Intention is conceptualized as a motivational construct reflecting the degree of effort an individual is prepared to invest in pursuing a target behavior [13,14,15,16,17]. Primary and meta-analytic studies indicate that intentions account for 20% to 40% of variance in behavior [13,16,17]. Nevertheless, there is an observed shortfall in the number of individuals who cite an intention to perform a behavior in the future and their subsequent enactment of the behavior [18]. This is indicated by the modest effect size for the intention–behavior relationship across studies [19]. For example, studies indicate that a substantive minority of individuals fail to act on their intentions [16,20]. This is often referred to as the intention–behavior “gap” [16,21,22]. Based on dual-phase theories, planning strategies that operate in a post-decisional manner after intention formation have been proposed as potential means to address this gap, e.g., [23].

### 1.2. Planning and Implementation Intentions

There is mounting evidence that planning strategies are effective in promoting successful enactment of intentions and, therefore, serve to moderate the intention–behavior relationship [6,16,20]. A planning strategy that has received a significant amount of attention is implementation intentions, a strategy in which individuals explicitly state a link between a critical internal or external cue or event and an intended goal-directed behavior [5,7,8,20,24]. Implementation intentions are often termed “if-then” plans and aim to facilitate a mental link between a critical situation, prompt, or cue and the intended behavior [5,25].

Implementation intentions are proposed to promote better intention enactment by facilitating the accessibility of the mental representation of the intended behavior when the situational cue specified in the plan is presented or becomes salient [16,24,26]. The act of planning, therefore, enables rapid and efficient execution of the intended behavior in response to the presentation of the linked cue, similar to habits [5,27,28]. It is also important to note that implementation intentions promote more efficient initiation of a specific goal-directed response to the critical situation, rather than facilitating motivation and intentions to pursue the goal. Research has supported the effects of implementation intentions in facilitating behavioral enactment and promoting stronger intention–behavior relations in various domains, e.g., [2,6,8,29,30,31,32,33,34]. In addition, research has also supported the mechanisms by which implementation intentions are proposed to promote better intention enactment, including better recall of the intended behavior [16] and more efficient, automatic behavioral enactment [27].

### 1.3. Planning as a Moderator of Implementation Effects

Although research has indicated consistent effects of implementation intention interventions in promoting behavioral enactment, and evidence that such interventions are effective in facilitating better intention enactment, there is still considerable unresolved variability in the effects of implementation intentions across studies. One issue that has received relatively little attention in the research literature is the conditions within the individual that may affect intention enactment and the effectiveness of planning interventions. An important observation in research on intention–behavior relations is that a substantive minority of individuals are effective in implementing their intentions without the need for a prompt to adopt a planning strategy, such as implementation intentions [18]. A likely explanation is that these individuals may possess greater self-regulatory skills or capacity to form plans unprompted, facilitating intention enactment. It follows, therefore, that individuals possessing such skills or capacity may benefit less when prompted to form implementation intention interventions, while those without may benefit considerably in terms of their intention enactment. Identifying constructs that confer the intuitive utilization of planning skills may, therefore, moderate the effectiveness of implementation intention interventions and provide useful information to those developing implementation intention interventions on whom may benefit the most in terms of intention enactment and behavior change.

Recently, researchers have examined the role of planfulness as a potential individual difference construct that may confer better self-regulation capacity, and we propose that it may be a candidate moderator of implementation intention interventions. Planfulness is defined as an individual’s tendency to adopt efficient goal-related cognitions and strategies in pursuit of goals [11]. Descriptors include “organized” and “methodical,” and individuals are said to vary on a continuum of planfulness, where higher scores indicate increased employment of effective cognitive processes and successful goal attainment. Planfulness has been classified as an underlying personality facet within the broader conscientiousness domain [12,35,36]. Ludwig and colleagues [11] constructed a self-report questionnaire measuring planfulness and identified three planfulness sub-facets: temporal orientation, mental flexibility, and cognitive strategies.

Research has supported the predictive validity of planfulness in promoting goal attainment and behavioral enactment [11,12,37]. For example, planfulness has been associated with the formation of plans on physical activity participation [37]. Individuals reporting high planfulness had stronger effects of planning intentions on their planning behavior than those reporting low or moderate planfulness. A more recent study demonstrated that planfulness predicted goal progress longitudinally [11]. Participants reporting higher planfulness indicated greater progress toward goal attainment than those reporting low planfulness. A further study found a positive association between planfulness and gym attendance, concluding that planfulness was associated with physical activity goal progress [12]. Overall, these studies demonstrate that planfulness moderates the intention–behavior relationship and predicts motivation and goal progress. These data provide preliminary evidence that individual differences in the capacity to plan may influence progress toward goals and serve to moderate individuals’ intention enactment.

### 1.4. The Current Research

The current study aimed to advance knowledge by examining whether individual differences in the capacity to plan moderates the effect of implementation intention interventions on behavioral enactment. The research builds on previous research indicating that some individuals manage to act on their intentions in the absence of planning interventions, suggesting that they have better capacity to augment their intentions with plans to enact them. We propose that this may be attributable to individual differences in planfulness, such that those reporting higher levels of planfulness exhibit better planning capacity. Such individuals are unlikely to benefit substantially from prompts to form an implementation intention because they probably already have sufficient propensity to form plans. By comparison, those with low planfulness are likely to benefit more because they have a specific deficit in their planning capacity. Therefore, we propose that implementation intention exercises may be more effective in promoting better behavioral participation and intention enactment among those reporting lower planfulness. We tested this hypothesis in an experimental study in which we examined the effect of an implementation intention intervention on individuals’ performance on a computerized go no-go task based on Brandstatter and colleagues [5] original study with individual differences in planfulness as a moderator.

While we expected better go no-go task performance for individuals who were prompted to form an implementation intention relative to those who were not, based on Brandstatter and colleagues’ observations, we expected this effect to be confined to those reporting low levels of planfulness. Specifically, we expected that individuals reporting low levels of planfulness would be more effective in plan enactment when prompted to form an implementation intention relative to individuals not prompted to form an implementation intention (H1). We also predicted that individuals reporting high planfulness would be equally effective in enacting their plans whether or not they formed an implementation intention (H2). We therefore expected an interaction between the implementation intention exercise and levels of planfulness. We expected the research to provide further insight into the factors that underpin goal-directed behavior and, in particular, the observed “gap” between intentions and subsequent action. All hypotheses were preregistered on the Open Science Framework (OSF) at https://osf.io/ytb37 (accessed on 21 May 2018).

## 2. Materials and Methods

### 2.1. Participants

Participants were undergraduate first-year students recruited from the participant pools from two universities. Students in each pool were asked to complete an initial screening questionnaire containing three planfulness questions taken from Ludwig and colleagues’ [11] scale. Equal numbers of participants with low (range = 1.00 to 3.49) and high (range = 4.10 to 5.00) averaged scores on the three items were invited to participate in the study. The low and high categories were characterized by the upper and lower thirds of the distribution of scores on the screening planfulness items across the two pools. First-year undergraduate psychology students were granted course credit for their participation. All other undergraduate students not eligible for course credit received a voucher for a free coffee.

A statistical power analysis using planfulness as a continuous variable and implementation intention condition as a binary variable with an interaction term was conducted using the Webpower application to estimate our target sample size. Although effects of implementation intentions have shown to be medium in size based on a meta-analysis of implementation intentions [2], we assumed a conservative small-sized effects for the planfulness × implementation intention effect in the current study. We therefore assumed that the interaction would account for an additional 5% of the variance in behavior beyond implementation intention and planfulness effects (assumed to be medium in size, 25%)—a R-squared change of 0.05. In addition, we set the alpha and the statistical power at 0.05 and 0.80, respectively. The analysis yielded a projected sample size of 112 participants. Note that our pre-registered sample size was 200 participants based on a statistical power analysis using an ANCOVA model with a medium-sized effect, power set at 0.80, alpha at 0.05, one covariate, and a tertile split (upper and lower 33.3 percentiles) of the planfulness variable. However, due to difficulties in recruiting sufficient participants with low planfulness, we fell short of our pre-registered sample size for a tertile split. On the advice of a statistician, we altered our analytic strategy to use a linear regression analysis, which had enabled us to use planfulness as a continuous variable, which not only yielded sufficient statistical power to detect a conservative small-sized effect size but also allowed us to use all of the available data. Our analysis reported here, therefore, deviates from the pre-registered analysis on this point.

### 2.2. Design

The study adopted an experimental design with two independent variables. The first independent variable was an experimental manipulation of planning. Participants were either prompted to form an implementation intention (“implementation intention” condition) or to merely familiarize themselves with the task (control or “familiarization” condition). Participants were randomly assigned to either the implementation intention or familiarization condition. Individual differences in planfulness constituted the second independent variable. The university laboratory was used as a covariate, to determine whether recruitment from the two universities affected outcomes. The dependent variable was reaction time on a computerized go–no-go task. Data files and scripts used to generate the analyses are available online: https://osf.io/ytb37 (accessed on 18 January 2022).

### 2.3. Procedure

Following completion of the screening questionnaire, eligible students recruited to participate in the main study were tested individually in laboratory conditions. On arrival to the laboratory, participants were welcomed by the experimenter, shown into an experimental cubicle, and asked to sit behind a computer desk. Participants were provided with instructions introducing the experiment as a study investigating the influence of decision-making on response times. Participants were informed that the study involved a computer task and questionnaire, as well as a brief pen-and-paper exercise, taking approximately 30 min to complete. Participants were then asked to read the information sheet and sign a consent form.

#### 2.3.1. Go No-Go Task

Next, participants received instructions for the study task, a go no-go task, which was administered using Inquisit experimental software and based on previous implementation intention experiments [36]. In the task, participants were presented with one of five numbers (1, 3, 5, 7, or 9) or capital letters (A, E, N, V, or X), presented in a random order on the screen for 1 second with an interstimulus interval of 1.5 s. A fixation appeared before each stimulus. Participants were required to respond to the numbers by pressing the “+” key and withhold responses to letters. A two-minute practice task was administered, and participant questions were addressed.

#### 2.3.2. Experimental Condition

Following the practice task, participants received the implementation intention or familiarization (control) manipulation. All participants were asked to draw a number from a shuffled deck of five cards, which would become their “critical number.” Participants were not aware that all cards were printed with the number 5. Participants randomly assigned to the implementation intention condition were prompted to fill in the blanks on a pen-and-paper implementation intention exercise, to make a plan to respond faster to their critical number. Participants were required to complete the plan, recite it five times, test their recall of the plan with the experimenter, and then repeat it again five times. If the plans were not recalled correctly, the planning task was repeated.

Participants assigned to the control (familiarization) condition were required to familiarize themselves with their critical number. This involved a pen-and-paper task in which participants wrote down their critical number a total of five times. All participants then completed a two-item manipulation check questionnaire. All participants then proceeded to the main trial of the go no-go task, which lasted 7 min.

#### 2.3.3. Questionnaire

Finally, participants were asked to complete an online planfulness measure administered by Qualtrics. All participants were given a unique identifier to use on the go no-go task, the pen-and-paper task, and the online questionnaire.

### 2.4. Measures

#### 2.4.1. Screening Questionnaire

A total of three screening questions were chosen from Ludwig and colleagues’ [11] planfulness measure, to obtain an estimated planfulness level. Participants were chosen to participate if they scored relatively low (average planfulness score between 1 and 3.49) or relatively high (average score between 4.1 and 5). The questions were: “I have a good sense of how I can work towards my long-term goals in the present,” “Developing a clear plan when I have a goal is important to me,” and “I think about my goal when I encounter obstacles to achieving it.”

#### 2.4.2. Demographic Information

Questions regarding the participant’s age, gender, marital status, educational achievement, employment status, income, and ethnicity were collected and used to describe the characteristics of the sample.

#### 2.4.3. Go No-Go Task

The computer-based go no-go task, as used in previous implementation intention research [38], was administered to participants using the Millisecond Inquisit 4 WebTM online research software. Participants were presented with a series of numeric (1, 3, 5, 7, and 9) and letter (A, E, N, V, and X) stimuli, which appeared in the center of the computer screen. Each stimulus number or letter was 2 cm in height and appeared for one second with an inter-stimulus interval of 1.5 seconds. A fixation point appeared before each stimulus. Each stimulus appeared in a randomized order across the practice and main trials, and across participants. Participants were required to respond to the number stimuli (targets) by pressing the “+” key on the keyboard numeric pad and inhibit responses to letter stimuli (distractors). Reaction times were measured for target and distractor stimuli. Better performance was indicated by faster (lower) reaction times on “go” trials of the task. The task involved one two-minute practice block and one seven-minute main trial block.

#### 2.4.4. Manipulation Checks

Two manipulation check items were used to measure participants’ confidence and intention towards completing the go no-go task with a focus on their critical number. One item was used to measure self-efficacy, “I am confident that I can respond quickly to the number five.” One item was used to measure intention, “I intend to respond as quickly as possible to the number 5.” These items will be scored using a seven-point Likert scale from strongly agree to strongly disagree. These items have been used in various studies and have been constructed from Ajzen’s Theory of Planned Behaviour research [39].

#### 2.4.5. Planfulness

Planfulness was assessed using Ludwig and colleagues’ [11] planfulness measure. The measure comprises 30 self-report items, rated on a five-point Likert scale (1 = strongly disagree to 5 = strongly agree). The scale consists of three 10-item subscales: temporal orientation, cognitive strategies, and mental flexibility. Half of the items in each subscale are reverse coded.

### 2.5. Data Analysis

We tested our hypotheses using moderated linear regression analysis. Specifically, reaction time from the go no-go task was regressed on to the main and interactive effects of planfulness as a continuous independent variable and implementation intention condition as a dummy-coded dichotomous variable. Planfulness was mean-centered prior to computing the planfulness × condition interaction term. Lab as a dummy-coded dichotomous variable was also included in the model as a covariate. In the event of a statistically significant interaction effect, we planned simple slopes follow-up analyses to examine planfulness effects on reaction time under the implementation intention and familiarization conditions.

## 3. Results

### 3.1. Participants

Data screening identified fives cases with missing reaction time data, two cases with incomplete planning manipulations, one outlier on the go no-go task, and one outlier on the planfulness scale. Following exclusion of these cases, the final sample comprised 208 participants (72% female). Participant ages ranged from 16 to 58 years (*M* = 22.77; *SD* = 7.23). The sample was predominantly Caucasian (67.8%), full-time students (47%) who have never been married (90.4%) and who reported an income of between USD 37,001 to USD 80,000 (30.8%). Participants also exhibited reasonable range about the mean on the planfulness measure in both the implementation intention (*M* = 3.55, *SD* = 0.52, range = 2.17) and familiarization (*M* = 3.55, *SD* = 0.55, range = 2.63) conditions.

### 3.2. Manipulation Check

Through a manipulation check, it was identified that there was no significant difference between the implementation intention and control conditions on intention *t*(206) = 0.27, *p* = 0.790, or self-efficacy *t*(206) = 1.64, *p* = 0.103. Participants did not differ in their self-efficacy or intention to respond to their critical number whether they received the implementation intention or familiarization instruction.

### 3.3. Regression Analyses

Preliminary examination of the histogram and scatterplots of residuals suggested that the assumptions of normality and homoscedasticity were met. All residuals were centered around zero with no significant skew, and consistent spread through the distributions indicated a normally distributed linear relationship. Again, individual skews of the planfulness and planning condition variables were not significant (did not exceed the SE ratio of 3.29 in absolute value). However, the individual skew for the dependent variable of the reaction time was skewed. A logarithmic transformation of the reaction time did not affect the findings, and thus the original variable was used for the analyses.

Zero-order correlations were calculated for all study measures. The correlation matrix is presented in Table 1. The predictor variables, planfulness, and implementation intention condition did not have a significant correlation with the reaction time.

The overall regression model was not significant, R^2^ = 0.005, *F*(4, 207) = 0.257, *p* = 0.905. Controlling for the effects of lab, neither planfulness, *b* = 6.28, *SE* = 8.36, 95% CI [−10.20, 22.76], β = 0.171, *t*(207) = 0.75, *p* = 0.453, nor the planning condition, *b* = 2.68, *SE* = 5.08, 95% CI [−7.33, 12.69], β = 0.037, *t*(207) = 0.53, *p* = 0.598, significantly predicted the reaction time. Importantly, the interaction between the planfulness and the condition, the key test for our experimental hypothesis, was also not statistically significant, *b* = −3.46, *SE* = 5.13, 95% CI [−13.57, 6.65], β = −0.154, *t*(207) = −0.68, *p* = 0.500. This led us to reject our hypotheses, and we abandoned the planned simple slopes analyses. Simple slopes for the effect of planfulness on go no-go task performance within each experimental group is illustrated in the Appendix A.

## 4. Discussion

The current study examined whether individual differences in planfulness moderated the effects of an implementation intention intervention on goal-directed behavior. We tested our predictions in an experimental study examining the effects of individual differences in planfulness and an implementation intention manipulation on performance on a computerized go no-go task used in previous implementation intention studies [5]. We predicted that participants reporting low levels of planfulness would report much greater effects of implementation intentions on reaction time on the go no-go task than those reporting high planfulness and that those reporting higher planfulness would not differ in the effects of implementation intentions on task performance. The results revealed no main or interaction effects of implementation intention manipulation and planfulness on task reaction times. The current results do not provide support for the moderating effect of planfulness on implementation effect, and it should be noted that we also failed to replicate Brandstatter et al.’s implementation intention effects. This research extends current knowledge by testing a key hypothesis relating the intra-individual factors that may lead to more effective planning and how such deficits may be addressed by planning interventions. That we could not promote better performance through planning, suggests that planning interventions in this context may not be as effective as previous studies suggest (e.g., Brandstatter et al., 2001). However, it should be stressed that there are a number of caveats relating to the methods in the current research, such as the use of a newly developed measure and a student sample with narrow, homogenous characteristics, that mitigate generalizability of the current findings and warrants subsequent revision.

A key finding of the current study was a failure to replicate the implementation intention effect observed in Brandstatter and colleagues’ original study, and findings are at odds with the extensive research that has supported implementation intention effects on goal-directed behavior in various populations and behaviors, e.g., [2]. However, it must be stressed that meta-analytic evidence on implementation intentions has generally been derived from field experiments and interventions rather than laboratory tests. Nevertheless, research examining implementation intention effects on laboratory task performance has also seen consistent positive effects beyond Brandstatter et al.’s experiment [24].

In the current study, we expected that the main effect of implementation intentions should be qualified by the interaction of the implementation intention effect with planfulness, but we found no interaction effect either. In fact, there were no correlations between any of the variables entered into the regression analyses. How might these null findings be interpreted? Certainly, they provide evidence that casts doubt on the proposed moderating effect of planfulness on the effect of implementation intentions on task performance, as well as some data that do not lend support to the implementation intention effect on task performance. However, it is important to put these findings into context and speculate on some potential caveats and moderators that may explain these findings that warrant further investigation. In particular, key methodological differences, particularly in the conduct of the tasks used, may have mitigated these effects, and we discuss these next.

There were some minor, but potentially important, methodological differences in the implementation intention tasks between the current study and Brandstatter et al.’s study. In the current study, we increased the rigor of the planning activity by involving additional repetitions and plan recall, whereas Brandstatter and colleagues provided a very simple instruction asking participants to tell themselves “if the number appears, I will respond particularly fast,” without elaboration or specifying how participants should reinforce this statement. We surmised that the original instruction may have been insufficient to engage participants in the planning exercise, so we introduced the elaborated version. This was the case for both the implementation intention and familiarization groups. With hindsight, this change may have impacted our findings: the familiarization task may have been sufficient to increase skills to manage the task thereby narrowing the differences in reaction time across the groups.

In addition, it is possible that participants were equally motivated to comply with the task in both implementation intention and familiarization conditions, as both conditions were instructed to respond quicker to the critical number; the difference lied in the strategy used to reinforce this critical number. Thus, individuals participating in the familiarization condition may have had a strong motivation to comply with the task despite the method of reinforcement. Thus, all participants may have simply complied with the computer task instruction rather than the planning activity [5].

Another possible mitigating factor was the difference in time taken on the manipulations in the implementation intention and familiarization groups. The implementation intention task took a total of five to ten minutes longer for participants to complete on average compared to the time taken for participants to complete the familiarization task. The length of time may, in retrospect, have been counterproductive in that the instructions demanded a higher degree of attention, and placed a higher cognitive load, on participants in the implementation intention condition compared to those in the familiarization condition. This may, ironically, have induced a greater level of fatigue among participants in this condition and may have worked against any behavioral advantage afforded by the formation of an implementation intention. This, however, cannot be verified as we did not collect data on fatigue in the current study, and it remains an open question for future research.

Third, inconsistencies between the results of the current study and previous research on implementation intentions may be due to participants’ lack of initial intention to perform the behavioral task and participation to simply receive course credit. Intentions are an important factor in predicting behavioral performance, and researchers [16,18] have demonstrated that intentions are the most proximal antecedent to behavior. Intentions are the attitudes and motivations towards a behavior; thus, if an individual does not possess an initial intention to perform a behavior, the likelihood of performance decreases [7,9,15,40]. In addition, self-determination theory (SDT [41]) highlights the facilitation of behavior through autonomy. SDT proposes that behavior change is more likely, effective, and endures over time if individuals are autonomously motivated [41]. In the current study, participants may not have been autonomously motivated to form intentions and were likely participating for extrinsic purposes, such as receiving course credit.

Finally, research has demonstrated that the effects of implementation intentions are more pronounced for tasks that are more difficult [5]. The greater the complexity of the task, the more effective the formation of implementation intentions is for goal achievement. However, when a task is considered easier, the benefit of implementation intentions is less pronounced. It is possible that the go no-go task was too simple, mitigating the effects of the implementation intention. It is also possible that participants did not fully engage with the implementation intention instructions.

### Strengths, Limitations, and Future Directions

The current study adopted a rigorous experimental design to examine the interactive effects of implementation intentions and planfulness on task performance. Specifically, we adopted the tasks and implementation intention manipulation exercises from a previously published study on implementation intentions, screened participants for planfulness to ensure we had a sufficient range of planfulness scores, and applied rigorous randomization, data collection, and analytic procedures. These strengths notwithstanding, other than the caveats relating to task differences between the current study and Brandstatter et al.’s original study, the current findings should be interpreted in light of several additional limitations. First, the measure of planfulness adopted is in its developmental stages, and while its construct, concurrent, and predictive validity, and internal consistency, have been noted [12], these findings need further corroboration. Second, the sample was comprised exclusively of university students, which means the findings should not be generalized to the broader population. Future research should aim to replicate findings in a non-student sample, preferably recruited using a random, stratified procedure. Third, we did not include measures of potential moderating factors such as intention, self-monitoring, and impulsivity. Future studies should aim to measure these constructs given their association with behavior and the potential to interact with planning strategies [1]. In addition, previous research suggests that individuals who exhibit good planfulness are more effective in achieving long-term goal progress [11]. Although planning for a short-term goal was not effective in bringing about changes in task performance in the current study despite every effort to replicate methods used in previous research [24], it may be that studying task performance over a longer period or identifying a suitably meaningful long-term goal would prove to be more effective as a dependent variable and a more robust test of our hypotheses.

## 5. Conclusions

We set out to test the hypothesis that a planning strategy, implementation intention, was likely more effective among individuals with lower capacity to engage planning, as determined by scores on the planfulness individual difference construct, than those with adequate capacity. The study was aimed to contribute to extending existing research on the effect of implementation intentions on behavior [5,6,9,20,24,42], by investigating the moderating effects of a within-person individual difference. The findings did not support our predictions, leading us to reject our hypotheses, and we were also not able to replicate the implementation intention effect. However, it should be stressed that the current study represents one data point amid the relatively consistent effects of implementation intentions on behavior across studies. We look to future research to address the need to systematically replicate current findings, directly as well as under varying conditions, such as the subtle, but potentially important, methodological variations with the original implementation intention manipulation and tasks adopted.

## Figures and Tables

**Table 1 behavsci-12-00047-t001:** Descriptive statistics and zero-order correlations between reaction time, condition, and planfulness.

Variables	1	2	3	M(SD)
1. Reaction time	-			30.63 (36.70)
2. Planfulness	0.021	-		3.54 (0.54)
3. Planning condition	0.036	−0.020	-	-

## Data Availability

Data files and scripts used to generate analyses are available online: https://osf.io/veqxt/ (accessed on 19 January 2022).

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
