# Peer review of "“The Best Laid Plans”: Do Individual Differences in Planfulness Moderate Effects of Implementation Intention Interventions?"

_behavsci, 2022, doi:10.3390/bs12020047_

Round 1

Reviewer 1 Report

The authors present a study that examined the influence of participant planfulness orientation (high/low) on their reaction time in a go/no-go task. The study presents an interesting approach in examining how our internal orientation for planning to achieve goals influences how we respond to a novel task. The presentation of the argument and the framing of the background information was well written. However, there are some issues in the methods, results, and discussion that need to be addressed.

Methods

The methods need to be presented in the order they were implemented (ie, the screening questionnaire is described last in the methods but this was part of participant selection). Overall, the section was difficult to follow at times and need to be revised for clarity.   

The authors present the course credit incentive as a potential issue in the discussion. The distribution of incentive needs to be clarified since some participants received course credit and others a gift card. How was it determined who would receive each incentive?

Were participants randomized? They were identified as either high or low for planfulness, but it does not state if they were placed randomly into one of the two conditions.

Define go no-go task performance.

Be consistent with how the “go no-go task” is presented in the text. It is presented in three different forms “go no-go,” “go/no-go”, and “go-no go”.

The description of the regression analyses should be in the methods section. This is a part of the research process, not the results of the study.

Discussion:

The discussion could be expanded, there are other ways that this study expands the literature. There are also other potential limitations. For instance, research cited in the background section note that individuals who exhibit good planfulness are more effective at achieving long-term goal progress. This study represented a short-term goal, how does this factor into the authors interpretation of results?

Other notes:

Lines 294-296 – Only the average for the planfulness measure is provided. However, there are 3 subscales mention in the methods (lines 252-254). Were there differences between the experimental and control based on the subscales? Also the hypotheses a focused on the High and low scoring for planfulness, how many of each participated in the two conditions?

Line 321 – Table 2 is referenced but there is not a table attached to the manuscript. There is also no mention of a Table 1 in the manuscript.

Lines 412 – 414 What does, “did not fully engage with the implementation intention instructions” mean? The participants did not complete the experimental task as instructed?

Author Response

REVIEWER’S COMMENT: The authors present a study that examined the influence of participant planfulness orientation (high/low) on their reaction time in a go/no-go task. The study presents an interesting approach in examining how our internal orientation for planning to achieve goals influences how we respond to a novel task. The presentation of the argument and the framing of the background information was well written. However, there are some issues in the methods, results, and discussion that need to be addressed.

AUTHORS’ RESPONSE: We thank Reviewer 1 for their positive feedback, and for taking the time to review our manuscript and provide feedback. We have addressed each of their comments below.

REVIEWER’S COMMENT: Methods: The methods need to be presented in the order they were implemented (ie, the screening questionnaire is described last in the methods but this was part of participant selection). Overall, the section was difficult to follow at times and need to be revised for clarity.

AUTHORS’ RESPONSE: The Methods section is now presented in the order of how each part of the experiment was implemented. Specifically, the screening questionnaire has been moved to the beginning of the measures section to ensure that measures used are presented in the order that they were completed by participants. The planfulness measure has been moved to the end of the measures section, as this was completed last by participants. Additional subheadings have been added to the section describing the procedure to improve clarity. The screening process is mentioned twice in the methods section to serve separate purposes; at the beginning in the participants section to describe the composition of the sample, and again in the measures section to describe how participants were recruited.

REVIEWER’S COMMENT: The authors present the course credit incentive as a potential issue in the discussion. The distribution of incentive needs to be clarified since some participants received course credit and others a gift card. How was it determined who would receive each incentive?

AUTHORS’ RESPONSE: First year psychology students were eligible for course credit, other (non-psychology) undergraduate students received a coffee voucher. We agree that this section was unclear, so we have re-written line 203 on page 4 of the Participants section as follows: “First year undergraduate psychology students were granted course credit for their participation. All other undergraduate university students not eligible for course credit received a free coffee voucher.”

REVIEWER’S COMMENT: Were participants randomized? They were identified as either high or low for planfulness, but it does not state if they were placed randomly into one of the two conditions.

AUTHORS’ RESPONSE: Yes, participants were randomized to the conditions within the planning intervention. Specifically, on line 228 of the manuscript we originally stated: “Participants randomly assigned to the implementation intention condition were prompted to fill in the blanks on a pen and paper implementation intention exercise, to make a plan to respond faster to their critical number.” We acknowledge that this may be unclear, so we have added in a sentence in the design section to more clearly state this. “Participants were randomly assigned to either the implementation intention or familiarization condition.” Please see page 5, line 226 of the revised manuscript.

REVIEWER’S COMMENT: Define go no-go task performance.

AUTHORS’ RESPONSE: Go no-go task performance has been clarified in the measures section describing the computer task. The following sentence has been added “Better performance is indicated by faster (lower) reaction times on ‘go’ trials of the task.”. Please see page 6, line 305 of the revised manuscript.

REVIEWER’S COMMENT: Be consistent with how the “go no-go task” is presented in the text. It is presented in three different forms “go no-go,” “go/no-go”, and “go-no go”.

AUTHORS’ RESPONSE: All presentations of the phrase have now been changed to “go no-go”.

REVIEWER’S COMMENT: The description of the regression analyses should be in the methods section. This is a part of the research process, not the results of the study.

AUTHORS’ RESPONSE: The description of the regression analysis has been moved to the end of the Methods section, under the sub-title “data analysis” as suggested (please see section 2.5 on pages 6-7 of the revised manuscript).

REVIEWER’S COMMENT: Discussion: The discussion could be expanded, there are other ways that this study expands the literature. There are also other potential limitations. For instance, research cited in the background section note that individuals who exhibit good planfulness are more effective at achieving long-term goal progress. This study represented a short-term goal, how does this factor into the authors interpretation of results?

AUTHORS’ RESPONSE: The contribution of this research to the literature has been further explained at the beginning of the Discussion section: “This research extends current knowledge by testing a key hypothesis relating the intra-individual factors that may lead to more effective planning and how such deficits may be addressed by planning interventions. That we could not promote better performance through planning, suggests that planning interventions in this context may not be as effective as previous studies suggest (e.g. Brandstatter et al., 2001). However, it should be stressed that there are a number of caveats relating to the methods in the current research, such as the use of a newly developed measure and a student sample with narrow, homogenous characteristics that mitigate generalizability of the current findings and warrants subsequent revision.” (Please see page 8, lines 463-471 of the revised manuscript).

The limitations section has also been extended to include the following passage that refers to broader contribution:

“In addition, previous research suggests that individuals who exhibit good planfulness are more effective in achieving long-term goal progress [11]. Although planning for a short-term goal was not effective in bringing about changes in task performance in the current study despite every effort to replicate methods used in previous research [22], it may be that studying task performance over a longer period or identifying a suitably-meaningful long-term goal would prove to be more effective as a dependent variable and a more robust test of our hypotheses”. (Please see page 10, lines 569-576 of the revised manuscript).

REVIEWER’S COMMENT: Lines 294-296 – Only the average for the planfulness measure is provided. However, there are 3 subscales mention in the methods (lines 252-254). Were there differences between the experimental and control based on the subscales? Also the hypotheses a focused on the High and low scoring for planfulness, how many of each participated in the two conditions?

AUTHORS’ RESPONSE: Although there are separate subscales, these subscales correlate very strongly and researchers typically aggregate the scales into a total score for use in predictive analyses (Ludwig et al., 2018; 2019).

REVIEWER’S COMMENT: Line 321 – Table 2 is referenced but there is not a table attached to the manuscript. There is also no mention of a Table 1 in the manuscript.

AUTHORS’ RESPONSE: This was an oversight. We have now re-reinstated the Table into the manuscript and labelled it Table 1.

REVIEWER’S COMMENT: Lines 412 – 414 What does, “did not fully engage with the implementation intention instructions” mean? The participants did not complete the experimental task as instructed?

AUTHORS’ RESPONSE: The phrase “did not fully engage” refers to the possibility that participants did not sufficiently pay attention or put effort into following the instructions and task in the implementation intention condition, which may have contributed to the lack of observed differences in intention and self-efficacy across conditions. Given that we found no differences in intention and self-efficacy measures as manipulation checks, this is a possibility. We also included self-efficacy measures as an additional manipulation check – intention was the primary check but we also examined effects of self-efficacy given its link to motivation and task performance (Bandura, 1986). We now mention this in the Method (please see lines page 6, line 311) and Results (please see page 7, line 399) sections of the revised manuscript.

Reviewer 2 Report

65. The intention-behavior relationship - It might be better to indicate more references here.

181-182. Equal numbers of participants with low (range = 1 to 3.49) and high  (range = 4.1 to 5) scores on the scale based on the distribution of scores across the two pools were invited to participate in the study. - Do you have any criteria with low and high scores? What made you divide score on 4? 

Author Response

REVIEWER’S COMMENT: 65. The intention-behavior relationship - It might be better to indicate more references here.

AUTHORS’ RESPONSE: Two additional references have been added into this section. See track changes on revised manuscript, and the two references copied below. Additional in-text references have also been added in this section of the manuscript.

  1. Cooke, R., & French, D. P. (2008). How well do the theory of reasoned action and theory of planned behaviour predict intentions and attendance at screening programmes? A meta-analysis. Psychology & Health, 23(7), 745-765. https://doi.org/10.1080/08870440701544437

  1. McEachan, R.R.C., Conner, M., Taylor, N.J. and Lawton, R.J.2011. Prospective prediction of health-related behaviours with the theory of planned behaviour: A meta-analysis. Health Psychology Review, 5, 97–144. https://doi.org/10.1080/17437199.2010.521684

REVIEWER’S COMMENT: 181-182. Equal numbers of participants with low (range = 1 to 3.49) and high (range = 4.1 to 5) scores on the scale based on the distribution of scores across the two pools were invited to participate in the study. - Do you have any criteria with low and high scores? What made you divide score on 4?

AUTHORS’ RESPONSE: Low and high scores on the planfulness measure were based on the distribution of scores on the pre-study screening administration of this measure to eligible participants. Those with aggregate planfulness scores in upper and lower thirds of the distribution of scores were invited to participate. This has been clarified in the manuscript. “Equal numbers of participants with low (range = 1.00 to 3.49) and high (range = 4.10 to 5.00) averaged scores on the three items were invited to participate in the study. The low and high categories were characterized by the upper and lower thirds of the distribution of scores on the screening planfulness items across the two pools.”

Reviewer 3 Report

The present study aimed to investigate the role of individual differences in planning propensity on the effect of planning strategies on the intention-behaviour gap. Specifically, this study aimed to investigate the effect of planning propensity on the effectiveness of executive intention in goal-directed behaviour using an experimental design. Participants in the study were first-year undergraduate students recruited from a pool of participants at two universities.

Statistical analyses conducted by the authors proved to be statistically insignificant, resulting in the rejection of the hypotheses. 

This study is very interesting and despite the lack of statistically significant results it is important. The lack of significance of the results in the mainstream should prompt the authors to present the source results on the study groups even more accurately. 

I therefore request that the authors complete the manuscript according to the following comments:

1/ the table referred to by the authors in line 321 is missing

2/ I suggest describing the values obtained in the experiment with a separate table and perhaps graphs. 

3/ I suggest posting in the repository referred to by the authors the data they write about in the manuscript - currently, apart from the study description, there is nothing more there despite the description in the text "Data Availability Statement: Data files and scripts used to generate analyses are available online:https://osf.io/ytb37."

Once this data has been completed, the manuscript will in my opinion be suitable for publication

Author Response

REVIEWER’S COMMENT: The present study aimed to investigate the role of individual differences in planning propensity on the effect of planning strategies on the intention-behaviour gap. Specifically, this study aimed to investigate the effect of planning propensity on the effectiveness of executive intention in goal-directed behaviour using an experimental design. Participants in the study were first-year undergraduate students recruited from a pool of participants at two universities.

Statistical analyses conducted by the authors proved to be statistically insignificant, resulting in the rejection of the hypotheses.

This study is very interesting and despite the lack of statistically significant results it is important. The lack of significance of the results in the mainstream should prompt the authors to present the source results on the study groups even more accurately.

I therefore request that the authors complete the manuscript according to the following comments:

AUTHORS’ RESPONSE: We thank Reviewer 3 for their positive feedback, and for taking the time to review our manuscript and provide feedback.

REVIEWER’S COMMENT: 1/ the table referred to by the authors in line 321 is missing

AUTHORS’ RESPONSE: This was also a point raised by Reviewer 2. This was an oversight, and the table has now been reinstated (please see page 7 of the revised manuscript).

REVIEWER’S COMMENT: 2/ I suggest describing the values obtained in the experiment with a separate table and perhaps graphs.

AUTHORS’ RESPONSE: The table mentioned in the manuscript has now been added and it provides descriptive statistics and zero-order correlations between reaction time, condition and planfulness (page 7 of the revised manuscript). The regression analysis has also been illustrated in a graph as suggested by the Reviewer. Specifically, simple slopes for the effect of planfulness on go no-go task performance within each experimental group are illustrated in the Figure in Appendix A available in the supplemental materials online: https://osf.io/veqxt/.

REVIEWER’S COMMENT: 3/ I suggest posting in the repository referred to by the authors the data they write about in the manuscript - currently, apart from the study description, there is nothing more there despite the description in the text "Data Availability Statement: Data files and scripts used to generate analyses are available online: https://osf.io/ytb37."

AUTHORS’ RESPONSE: The data and output files have now been posted in the OSF repository as suggested.

REVIEWER’S COMMENT: Once this data has been completed, the manuscript will in my opinion be suitable for publication.

AUTHORS’ RESPONSE: Again, we want to thank Reviewer 3 for their feedback.

Round 2

Reviewer 1 Report

The author(s) did a nice job of responding to reviewers' recommendations and have overall improved their manuscript. I do not have any additional comments at this time. 

Reviewer 3 Report

Accept in present form